# Probabilistic Human Health Risk Assessment of Inorganic Arsenic Exposure Following the 2020 Taal Volcano Eruption, Batangas, Philippines

**DOI:** 10.3390/toxics14010013

**Published:** 2025-12-22

**Authors:** Yu-Syuan Luo, Jullian Patrick C. Azores, Rhodora M. Reyes, Geminn Louis C. Apostol

**Affiliations:** 1Master of Public Health Program, College of Public Health, National Taiwan University, Taipei 100025, Taiwan; 2Institute of Food Safety and Health, College of Public Health, National Taiwan University, Taipei 100025, Taiwan; 3Population Health Research Center, College of Public Health, National Taiwan University, Taipei 100025, Taiwan; 4Department of Physical Sciences and Mathematics, University of the Philippines Manila, Manila 1000, Philippines; 5East Avenue Medical Center-Toxicology Referral and Training Center (EAMC-TRTC), Disease Prevention & Control Bureau, Department of Health, Quezon 1100, Philippines; 6School of Medicine and Public Health, Ateneo de Manila University, Don Eugenio Lopez Sr. Medical Complex, Ortigas Ave, Pasig 1604, Philippines

**Keywords:** arsenic, food contamination, risk assessment, volcanic eruptions, Monte Carlo method

## Abstract

Volcanic eruptions can mobilize naturally occurring toxic elements such as arsenic into surrounding ecosystems, contaminating soil, water, and food webs. Despite increasing evidence of arsenic enrichment in volcanic regions, comprehensive exposure assessments that integrate dietary and drinking water pathways remain limited, particularly in post-eruption contexts where baseline data are scarce. Following the 2020 Taal Volcano eruption, this study conducted a probabilistic risk assessment to quantify aggregate exposure to inorganic arsenic (iAs) among residents of Batangas, Philippines. A Monte Carlo simulation framework (10,000 iterations) integrated post-eruption environmental data on total arsenic in soil, lake water, drinking water and clam tissues with modeled bioaccumulation and transfer factors for fish and major terrestrial crops. Two exposure scenarios, lower bound (50% iAs fraction) and upper bound (90% iAs fraction), were applied to capture uncertainty in arsenic speciation and bioavailability. Simulated iAs concentrations followed the order rice > corn > vegetables > root crops. Aggregate daily iAs doses averaged 3.0 ± 1.4 µg/kg bw/day (lower bound) and 4.0 ± 2.0 µg/kg bw/day (upper bound), with females showing slightly higher exposures and pregnant women exhibiting lower doses. Sensitivity analysis identified clam intake, rice intake, and iAs in rice, clams, and drinking water as dominant determinants of total exposure. All simulated individuals exceeded the U.S. EPA non-cancer reference dose (HQ > 1) and cancer risk benchmark (10^−6^–10^−4^), indicating substantial health concern. These findings highlight the urgent need for sustained environmental monitoring, arsenic speciation analyses, biomonitoring, and public health programs to guide evidence-based management in arsenic-affected volcanic regions.

## 1. Introduction

Volcanic eruptions represent significant environmental hazards, capable of mobilizing naturally occurring toxic elements, such as arsenic, into surrounding ecosystems through multiple pathways [1]. Volcanic deposits, particularly fine ash rich in glass shards, are recognized globally as important geogenic sources of arsenic in water, soil, and sediments [2,3]. During and after eruptions, arsenic dispersed through volcanic ash fallout can contaminate water supplies and agricultural soils, eventually entering the food chain via bioaccumulation in crops and aquatic organisms [4,5]. These contamination processes pose substantial public health risks, particularly in densely populated areas adjacent to active volcanic systems where communities depend heavily on local water and food resources.

The 2020 Taal Volcano eruption in Batangas, Philippines, exemplified these environmental threats, with the explosive event dispersing volcanic materials across surrounding communities. Our previous investigation documented elevated arsenic concentrations in groundwater sources in communities surrounding Taal Volcano, highlighting the immediate post-eruption contamination of drinking water resources [6]. While most previous studies have focused on environmental concentrations or hydrogeochemical processes [7,8,9], none have quantified population-level aggregate exposure or health risks using probabilistic methods in a post-eruption context. Arsenic mobilization into agricultural and aquatic systems may create sustained dietary exposure pathways [10], underscoring the need for a more integrated assessment.

Rice, a dietary staple throughout Southeast Asia including the Philippines, is particularly efficient at accumulating arsenic, with concentrations typically ten times higher than other cereal crops due to its cultivation under flooded conditions that favor arsenic mobilization and uptake through silicon transporters [11,12]. In arsenic-affected regions, rice consumption has been identified as a major dietary source of inorganic arsenic (iAs) exposure, contributing substantially to cardiovascular disease and cancer risk even in populations with relatively low drinking water arsenic concentrations [13,14,15,16]. Similarly, aquatic foods harvested from contaminated water bodies can contribute significantly to aggregate arsenic exposure, particularly in communities with high seafood consumption patterns characteristic of Southeast Asian populations [17,18,19,20]. Arsenic occurs in both organic and inorganic forms, with iAs being the more toxic species linked to carcinogenicity [21,22] and a range of non-cancer health effects [23,24,25]. Comprehensive exposure assessments for iAs are thus essential in volcanically impacted regions. However, measured iAs data across a wide range of food crops are typically limited, constraining the accuracy of exposure and risk estimates.

To address this gap, the present study provides the first probabilistic human health risk assessment integrating multiple exposure pathways, including drinking water, lake water, clams, rice, corn, vegetables, and root crops, following the 2020 Taal eruption. Using available arsenic measurements and population-specific dietary intake patterns, we estimated aggregate iAs exposure and characterized associated non-cancer and cancer risks. Sensitivity analyses were used to identify the most influential exposure contributors and demographic groups of concern.

## 2. Materials and Methods

### 2.1. Study Design

We conducted a probabilistic human health risk assessment to characterize aggregate iAs exposure among residents of Batangas following the 2020 Taal Volcano eruption. The assessment focused on the immediate to medium-term post-eruption period (2020–2022). Both dietary and drinking water pathways were included. A Monte Carlo simulation framework (10,000 iterations) was used to characterize variability in contaminant concentrations, food intake, and body weight. Two exposure scenarios were evaluated. For terrestrial crops, iAs was assumed to represent either 50% (lower-bound, LB) or 90% (upper-bound, UB) of total arsenic, based on typical ranges reported by EFSA for cereals and plant-based foods [26].

### 2.2. Environmental Arsenic Inputs

#### 2.2.1. Drinking Water

The arsenic concentrations in drinking water used for exposure modeling and risk assessment were obtained from a previously published field study [6] that assessed arsenic levels in groundwater sources across selected communities surrounding Taal Volcano, Batangas Province, Philippines. In brief, a total of 72 drinking water samples were analyzed from 26 wells in 11 municipalities and one city, collected in 2020 (*n* = 32) and 2021 (*n* = 41). Sampling locations included deep wells (*n* = 13), shallow wells (*n* = 4), protected springs (*n* = 1), and communal waterworks systems (Level III, *n* = 8), reflecting typical drinking water sources across the region. The sampled wells were geospatially distributed within the 7 km, 10 km, and 14 km volcanic danger zones defined by the Philippine Institute of Volcanology and Seismology (PHIVOLCS), as well as areas beyond this perimeter. The sample collection protocol, sample pretreatment, and determination of total arsenic levels in drinking water are detailed elsewhere [6]. Total arsenic concentrations in drinking water ranged from 0.000735 to 0.11 mg/L, which were used directly as drinking water inputs (Table 1).

#### 2.2.2. Total Arsenic Levels in Soil and Taal Lake Water

Total arsenic concentrations in soil and Taal Lake water were obtained from post-eruption environmental monitoring conducted after the January 2020 Taal Volcano eruption [7]. Soil samples (0–10 cm) from agricultural areas in Cuenca, Talisay, and Tagaytay contained 1.92–7.91 mg/kg total As. Lake water samples collected from three nearshore sites in Agoncillo, Cuenca, and Talisay (~500 m from shore) contained 0.005–0.0097 mg/L total As. These values served as inputs for reconstructing arsenic concentrations in terrestrial crops and aquatic foods.

#### 2.2.3. Simulation of iAs Concentrations in High-Risk Foods

##### Food Selection Rationale

Fish, clams, rice, corn, vegetables, and root crops were selected as representative food groups based on three criteria: (1) dietary relevance in Batangas and the broader Philippine population; (2) plausible pathways for arsenic transfer from contaminated water or soils into edible tissues; and (3) availability of consumption data from national nutrition surveys. Fish and clams were included because they integrate lake and groundwater inputs across local food webs [27]. Rice, vegetables, and root crops were selected due to their potential to accumulate arsenic from soil and irrigation water [26], and because they represent staple components of regional diets. Corn was included as an additional commonly consumed staple [28]. These food groups provided sufficient coverage of major dietary exposure pathways needed for probabilistic modeling.

##### Aquatic Foods

Fish. iAs concentrations in fish (mg/kg) were estimated using:Lake water total As × bioaccumulation factor (BAF = Uniform [10.3, 22] L·kg^−1^) [29]Seasonal multipliers to reflect dry-season concentration (TruncNorm mean = 6.71, SD = 4.50, min = 2.21, max = 13.66) [30]Fraction of total As present as iAs (Uniform [0.117, 0.142]) [29]

Clam. Empirical dry-weight total As concentrations (5.0–6.3 mg/kg) were taken from post-eruption measurements of *Corbicula fluminea* collected at three active harvesting sites [31]. Total As was modeled using a truncated normal distribution (mean = 5.67 mg/kg, SD = 3) (Appendix A). The same iAs fraction multiplier used for fish was applied. This parameterization reflects the contamination baseline of Taal Lake clams in the post-eruption period, providing site-specific realism for probabilistic exposure modeling.

##### Terrestrial Crops

Total As concentrations in crops (µg/g) were estimated by multiplying soil As levels (see Section 2.2.2) with crop-specific transfer factors drawn from literature-based ranges (Table 2) [32,33]. To account for variability in arsenic speciation and plant uptake, these total As concentrations were further multiplied by a fixed inorganic fraction—0.9 for the UB scenario and 0.5 for the LB scenario.

### 2.3. Exposure Factors and Population Parameters

#### 2.3.1. Drinking Water Intake and Food Consumption Data

Daily drinking water consumption was set at 1.79 L per person, based on the 2018–2019 Philippine Nutrition Facts and Figures: Food Consumption Survey conducted by the Department of Science and Technology–Food and Nutrition Research Institute (DOST-FNRI) [34]. Because variability estimates (e.g., standard deviation, SD) were not reported in the source dataset, this intake value was treated as a fixed parameter in the exposure model.

Daily intake values for fish, rice, corn, vegetables, and root crops were modeled using truncated normal distributions to represent inter-individual variability while enforcing biologically plausible bounds (Table 3). Mean and standard deviation values were obtained from the Philippine National Nutrition Survey [35,36]. Lower bounds were set at 0 g/day, and upper bounds were capped at the mean + 3 SD. Truncated distributions were generated using the rtruncnorm () function in R, and these probabilistic inputs were incorporated into the Monte Carlo simulation to estimate total and pathway-specific iAs intakes.

#### 2.3.2. Demographical Parameters

To incorporate population variability in physiological parameters relevant to dose normalization, we modeled sex distribution, body weight, and height for 10,000 simulated individuals using demographic data and literature-based assumptions. Sex distribution was assigned based on the 2020 Philippine Census, with a population-proportional split of 50.6% male and 49.4% female [37]. Body weight and height were modeled using sex-specific truncated normal distributions informed by Filipino anthropometric data [38] and supplemented by comparable regional data (e.g., Taiwanese population studies). All anthropometric distributions were generated using the rtruncnorm () function in R, ensuring simulated values remained within biologically reasonable bounds (Table 4). These parameters were subsequently used to compute individual-level intake doses (µg/kg-day) by normalizing estimated iAs intakes to body weight.

### 2.4. Aggregate Exposure Modeling of iAs

Aggregate exposure to iAs among the Batangas population was estimated by integrating exposure from dietary intake and drinking water ingestion using Monte Carlo simulation (10,000 iterations). For each simulated individual, the daily intake of iAs (µg/kg-day) was computed as the sum of contributions from all exposure media:$$Estimated\;daily\;iAs\;intake\;(E_{iAs})=\sum_{i}(C_{i}\times {IR}_{i})/BW$$
where *C*_*i*_ is the iAs concentration in medium *i* (µg/g or µg/L, see Section 2.2), and *IR_i_* is the corresponding individual consumption rate (g/day or L/day, see Section 2.3). *BW* (kg) represents the body weight for the simulated individuals. Two parallel exposure scenarios were generated to represent lower-bound and upper-bound conditions, capturing the uncertainty in iAs bioavailability from terrestrial foods.

### 2.5. Sensitivity Analysis

A global sensitivity analysis was performed to identify key parameters influencing daily iAs intake. Two complementary methods, Standardized Regression Coefficients (SRC) and Partial Rank Correlation Coefficients (PRCC), were applied using 10,000 Monte Carlo iterations. Eighteen input variables were evaluated, including environmental total As concentrations, bioaccumulation or transfer factors, the dry-season conversion factor, food consumption rates, and body weight. SRC and PRCC results were summarized using tornado plots to illustrate the relative influence of each parameter on exposure variability. Parameters with *p* < 0.05 were considered statistically significant.

### 2.6. Risk Characterization

Health risks associated with iAs exposure were characterized by estimating both non-cancer and cancer risk metrics based on the simulated daily exposure doses (µg/kg-day). For non-cancer effects, the hazard quotient (HQ) was calculated for each iteration as:$$HQ=\frac{E_{iAs}}{RfD}$$
where *E_iAs_* is the estimated daily iAs exposure dose, and *RfD* is the oral reference dose for inorganic arsenic (0.06 μg/kg-day) established by the U.S. Environmental Protection Agency [39]. An HQ greater than 1 indicates a potential for adverse health effects.

For cancer risk, lifetime excess cancer risk (ECR) was computed using:$$ECR=E_{iAs}\times CSF$$
where *CSF* is the oral cancer slope factor for inorganic arsenic (0.032 (μg/kg-day)^−1^). Cumulative probability distributions of HQ and ECR were generated to evaluate population-level variability and risk exceedance probabilities.

### 2.7. Data Analysis and Visualization

All analyses were performed in R (version 4.4.0) and RStudio (2023.06.01 Build 421, Posit Software, PBC) using packages dplyr (version 2.5.1), truncnorm (version 1.0–9), ppcor (version 1.1), and ggplot2 (version 4.0.0). The boxplots, pie chart, violin plots, cumulative density plots, and scatter plots were generated using GraphPad Prism 9 (version 9.5.1).

## 3. Results

### 3.1. Simulated Concentrations of iAs in Fish and Terrestrial Crops

iAs concentrations across fish and terrestrial crops were modeled based on the total As levels in soil and Taal lake water (Figure 1). Simulated iAs level in fish was 0.016 ± 0.005 μg/g during rainy season and 0.12 ± 0.06 μg/g during dry season. Among terrestrial crops, rice exhibited the highest iAs concentration, followed by corn, vegetables, and root crops. Under the UB scenario (assuming 90% of total arsenic as iAs), concentrations across crops ranged from 0.02–0.12 µg/g, whereas under the LB scenario (assuming 50% iAs), they ranged from 0.007–0.06 µg/g.

### 3.2. Contribution of Individual Exposure Pathways to iAs

The pie charts illustrate the relative contributions of different dietary and drinking water pathways to iAs exposure among residents of Batangas (Figure 2). Under the LB scenario (Figure 2A), clams represented the largest contributor to total iAs dose (26.2%), followed by rice (22.7%), fish (15.8%), drinking water (14.2%), vegetables (11.1%), corn (9.5%), and root crops (0.6%). Under the UB scenario (Figure 2B), rice became the dominant contributor (30.4%), followed by clams (19.3%), vegetables (14.4%), corn (12.7%), fish (11.8%), drinking water (10.7%), and root crops (0.8%). Across both scenarios, aquatic foods (fish and clams) accounted for approximately 31–42% of total iAs intake, while terrestrial crops (rice, corn, vegetables, and root crops) contributed 44–58%. These results indicate that dietary sources, particularly rice and aquatic foods, remain the primary exposure pathways for residents of Batangas.

### 3.3. Aggregate Dose of iAs Under Lower- and Upper-Bound Exposure Scenarios for Residents in Batangas, Philippines

The aggregate daily doses of iAs across the Batangas population were successfully modeled using the Monte Carlo framework (Figure 3). Overall, females showed slightly higher central tendencies than males. Pregnant women exhibited marginally lower aggregate iAs doses (LB: 1.8 ± 1.1 µg/kg bw/day; UB: 2.0 ± 1.1 µg/kg bw/day), although differences were not statistically significant as determined by one-way ANOVA.

### 3.4. Sensitivity Analysis of Input Exposure Variables

The global sensitivity analysis identified a consistent set of parameters that dominated variability in aggregate iAs intake across both LB and UB scenario (Figure 4). Under the LB scenario, clam intake, iAs concentration in drinking water, rice intake, and body weight were the strongest predictors of exposure, with similar patterns observed across both SRC and PRCC analyses. Concentrations of iAs in clams and rice, as well as consumption of fish, vegetables, and corn, also contributed meaningfully to model variability.

Under the UB scenario, rice-related parameters, including rice intake and iAs concentration in rice, emerged as the primary drivers of exposure, followed by body weight, clam intake, and iAs concentration in drinking water. Contributions from vegetable, corn, and fish pathways remained moderate but consistent across both sensitivity methods.

Across all analyses, a small subset of food and environmental parameters accounted for most of the total variability in modeled iAs intake, indicating that exposure was predominantly driven by a combination of key dietary staples and drinking water contamination patterns rather than the full set of input variables.

### 3.5. iAs Induced Non-Cancer and Cancer Risk

All simulated individuals exhibited HQs greater than 1 under both LB and UB exposure scenarios, indicating potential concern for non-cancer health effects associated with iAs exposure (Figure 5). The maximum HQs reached 212.8 in the LB scenario and 286.1 in the UB scenario, showing that combined dietary and drinking water exposures substantially exceeded the U.S. EPA reference dose (0.06 μg/kg/day).

For cancer risk, ECRs were estimated using simulated aggregate iAs doses and the oral cancer slope factor (Figure 6). All modeled individuals exceeded the acceptable benchmark of 10^−6^ for the general population, with ECRs under both scenarios primarily distributed between 10^−2^ and 1. These findings indicate a markedly elevated lifetime cancer risk among Batangas residents and underscore the need for continued monitoring and mitigation of iAs contamination in both aquatic and terrestrial food pathways.

## 4. Discussion

Volcanic activity can profoundly alter the geochemical processes in surrounding ecosystems, mobilizing naturally occurring elements such as arsenic into soil, water, and food chains. Following the 2020 Taal Volcano eruption, concerns emerged regarding elevated iAs exposure among nearby populations. Although arsenic contamination in volcanic regions has been documented globally, few studies have applied an aggregate, probabilistic approach integrating both dietary and drinking water pathways [26,40]. By combining environmental monitoring data with Monte Carlo–based exposure modeling, this study provides the first quantitative characterization of post-eruption iAs exposure among residents of Batangas, Philippines. The findings contribute to a better understanding of how volcanic events can intensify environmental contamination and potentially influence long-term population health risks.

Overall, the modeled iAs concentrations in food crops were consistent with measured values reported in the literature. The reconstructed concentrations of iAs in terrestrial crops followed the order rice > corn > vegetables > root crops, consistent with findings from the European Food Safety Authority [40]. EFSA similarly identified rice and rice-based products as the dominant contributors to dietary iAs intake due to the efficient uptake of arsenic under flooded paddy conditions, where reduced soil environments promote arsenite formation and bioavailability. The modeled iAs concentrations in this study align with real-world monitoring data, in which rice typically contains 0.026–0.76 mg/kg and vegetables and tubers generally remain below 0.03 mg/kg [41]. The mean iAs concentrations in rice (0.13 mg/kg) reported in the literature is within the same order of magnitude as the simulated values in this study.

Additional studies from Asia provide further support for the modeled trends. Field investigations in Bangladesh have reported rice iAs concentrations ranging from 0.29 to 0.51 mg/kg across regions with elevated soil or irrigation-water arsenic [42]. Similarly, surveys from Vietnam and Thailand have documented iAs levels in rice between 0.12 and 0.67 mg/kg [43,44], consistent with the modeled range. Concentrations measured in root crops and leafy vegetables in arsenic-impacted areas of China also generally fall below 0.03 mg/kg [45], matching the lower iAs uptake predicted for these food groups in our model.

Similarly, the modeled iAs concentrations in fish were comparable with field measurements from the Philippines. In Laguana de Bay, total As concentrations in fish ranged from 0.15–0.39 mg/kg during the dry season and 0.018–0.107 mg/kg during the wet season [30]. Assuming that 11.7–14.2% of total As occurs in the inorganic form, the modeled iAs concentrations in this study fall within the same order of magnitude. This correspondence suggests that the applied bioaccumulation and dry-season correction factors effectively represent the hydrological and geochemical dynamics of Taal Lake following the 2020 eruption. The strong agreement between simulated and observed data supports the reliability of the probabilistic reconstruction framework in characterizing post-eruption arsenic contamination in the Batangas region.

Although the majority of arsenic in fish and clams exists as less toxic organic forms, aquatic foods remain important contributors to aggregate iAs exposure. This is attributable to the combination of relatively high total arsenic concentrations in these species and the substantial aquatic food consumption rates among Filipinos (mean fish intake = 59 g/day; “crustaceans and molluscs,” representing clams = 8.5 g/day). Experimental evidence further indicates that under elevated iAs exposure, biotransformation efficiency in clams may decline, leading to increased accumulation of iAs [46]. In Asaphis violascens, exposure to rising concentrations of As(III) or As(V) resulted in a marked reduction in arsenobetaine (AsB) and a concomitant increase in dimethylarsinic acid (DMA) and iAs fractions—reaching up to several tens of percent under high exposure conditions. Uptake of As(III) was found to exceed that of As(V), suggesting dose-dependent bioaccumulation and altered speciation dynamics. These findings demonstrate that even when the inorganic fraction is relatively small, the combination of high total arsenic concentrations and frequent seafood consumption can substantially elevate aggregate exposures among lake-dependent communities.

Local reliance on Taal Lake fisheries increases vulnerability to arsenic exposure, particularly for households that depend on lake-derived aquatic foods. Our results identified clams and fish as major contributors to aggregate iAs intake, underscoring the importance of monitoring lake-sourced foods in post-eruption conditions. Effective risk management in these communities should prioritize continued biomonitoring, clear consumption guidance, and support for households whose livelihoods may be affected by contamination-related restrictions.

In this study, pregnant women exhibited lower aggregate iAs doses than the general adult population. While reduced consumption of fish and shellfish during pregnancy has been documented in parts of Southeast Asia [47], this cultural practice represents only one plausible explanation. Other contributing factors may include lower overall food intake during pregnancy, intentional dietary adjustments guided by prenatal health advice, and differences in preferred food items that carry lower arsenic burdens. Despite these lower exposures, pregnant women in this study still exceeded the U.S. EPA reference dose by more than 30-fold, underscoring the heightened developmental risks associated with prenatal exposure. Given established links between prenatal arsenic exposure and adverse outcomes, including reduced birth weight, impaired neurodevelopment, and increased infant mortality [25,48], targeted interventions remain essential, such as prenatal urinary arsenic monitoring, guidance on low-arsenic dietary options, and nutritional support to enhance arsenic methylation capacity.

The sensitivity analysis identified clam intake, rice intake, iAs in rice, iAs in clams, and iAs in drinking water as the dominant determinants of aggregate exposure. This finding aligns with well-established evidence that food and drinking water constitute the principal exposure pathways for iAs [49,50]. According to EFSA’s report, rice and rice-based products are consistently the largest contributors to dietary iAs exposure across all age groups, followed by drinking water and other grain-based commodities [40]. Rice is uniquely efficient in accumulating arsenic because its cultivation under flooded, anaerobic conditions enhances the mobilization and uptake of arsenite, the more bioavailable and toxic inorganic form [51]. Similarly, shellfish and bivalves such as clams can accumulate arsenic through sediment interactions, making them important but variable contributors to total intake depending on local geochemical conditions [52]. Drinking water, though generally containing lower concentrations (~2 µg/L on average), remains a critical exposure route due to high daily consumption volumes [53]. These findings mirror the present study’s results, in which both dietary and waterborne pathways were key determinants of aggregate exposure, underscoring the need for integrated mitigation strategies targeting both food and water contamination.

All simulated individuals exhibited HQs greater than 1 in both lower- and upper-bound scenarios, with maximum HQs of 212.8 and 286.1, respectively. These levels far exceed the U.S. EPA’s reference dose of 0.06 µg/kg bw/day, indicating substantial non-cancer health risks. Likewise, all simulated lifetime excess cancer risks exceeded the benchmark range of 10^−6^–10^−4^, with most values between 10^−2^ and 1, suggesting markedly elevated cancer risks. These risk magnitudes are comparable to those observed in volcanic or geothermal regions such as Latin-American countries [54] and southern Italy [55], where naturally occurring arsenic contamination has led to significant population-level health impacts. These results highlight the urgent need for systematic environmental and biomonitoring programs to assess exposure trends over time and evaluate the effectiveness of mitigation strategies. Furthermore, establishing post-eruption baseline levels of iAs in soil, water, and locally produced food is essential to accurately characterize long-term exposure and effectively communicate environmental health risks to affected communities.

Despite these important findings, several limitations should be acknowledged. First, this assessment used total arsenic measurements from soil, lake water, and clams, while iAs in locally grown crops and fish was not directly measured. Future work incorporating speciation analyses and expanded food monitoring would help reduce this uncertainty. Second, the transfer and bioaccumulation factors applied in the reconstruction models were derived from studies conducted in other geochemical settings and may not fully represent the volcanic soils of Batangas. Local soil properties, including post-eruption pH, mineral composition, and redox conditions, may influence arsenic mobility and crop uptake differently from regions where most transfer factor studies have been performed. Third, dietary patterns in lakeshore communities may differ from national averages—particularly for fish and clams—and vulnerable subgroups may have distinct consumption profiles and susceptibility; however, age-stratified food and drinking water intake data necessary to model these groups were not available, limiting our ability to perform subgroup-specific exposure assessments. Finally, the absence of pre-eruption baseline data limits our ability to assess temporal changes in environmental arsenic following the 2020 eruption. These gaps underscore the need for region-specific monitoring and validation in future studies.

Notwithstanding these limitations, this study provides the first integrated, post-eruption probabilistic assessment of iAs exposure in Batangas. The findings demonstrate that both dietary and drinking water pathways substantially contribute to elevated health risks among residents, particularly through rice, clams, and drinking water. These results underscore the urgent need for comprehensive monitoring, targeted risk communication, and sustainable mitigation strategies to protect public health. Establishing long-term surveillance and biomonitoring programs will be critical for tracking exposure trends and supporting evidence-based management in this and other volcanic regions affected by arsenic contamination.

## 5. Conclusions

This study provides the first probabilistic assessment of iAs exposure following the 2020 Taal Volcano eruption and shows that rice and lake-derived aquatic foods are the primary contributors to intake in Batangas. Although pregnant women had lower exposures than other adults, all groups exceeded health-based guidance values. Important uncertainties remain, including the absence of local iAs speciation data, region-specific transfer factors, and age-stratified consumption information. Addressing these gaps is essential for improving future assessments. The findings offer an evidence base to support arsenic monitoring, food safety management, and risk communication in communities affected by volcanic activity.

## Figures and Tables

**Figure 1 toxics-14-00013-f001:**
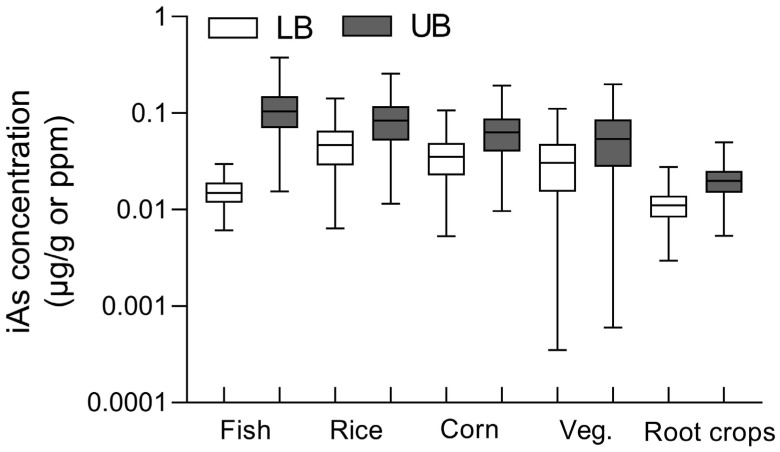
Simulated iAs concentrations in major food groups under lower-bound and upper-bound exposure scenarios. Boxplots show the estimated iAs distribution (µg/g or ppm, log scale) for fish, rice, corn, vegetables, and root crops. The lower-bound (LB; white boxes) and upper-bound (UB; gray boxes) scenarios represent conservative and worst-case assumptions, respectively, reflecting differences in seasonal concentration factors and conversion rates of total arsenic to inorganic arsenic. Boxes show medians and minimum-maximum ranges, with whiskers indicating 5th–95th percentiles.

**Figure 2 toxics-14-00013-f002:**
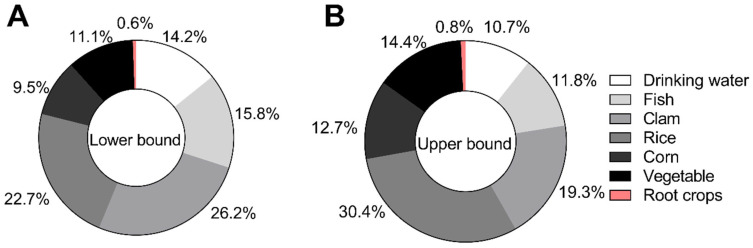
Contribution of individual exposure pathways to total iAs intake under lower-bound and upper-bound scenarios. (**A**) The left panel shows the percentage contribution of each pathway under the lower-bound scenario (iAs conversion factor = 0.5). (**B**) The right panel reflects the upper-bound scenario (iAs conversion factor = 0.9).

**Figure 3 toxics-14-00013-f003:**
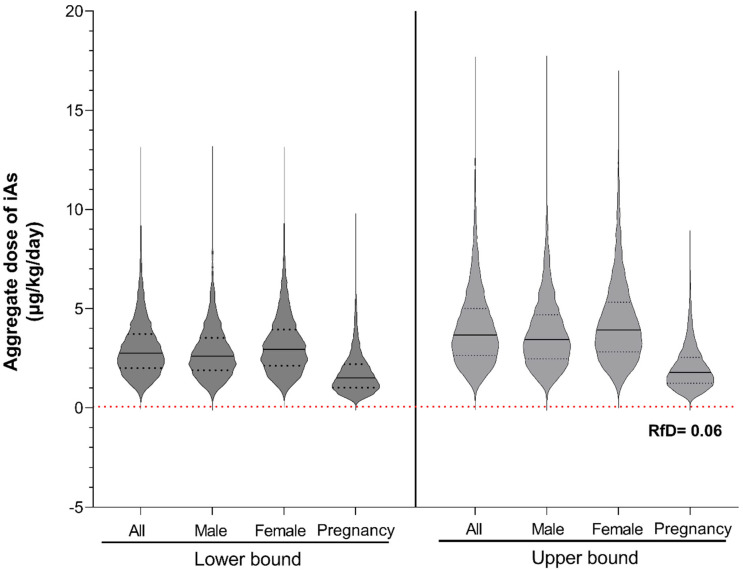
Distribution of simulated iAs dose under lower- and upper-bound exposure scenarios. Violin plots show the distribution of daily iAs doses (µg/kg-day) for the total population (All) and subgroups (Male, Female, Pregnancy). The left and right panels represent lower-bound and upper-bound scenarios, respectively. The black line denotes the median, while the black dash lines represent the interquartile range. The red dashed line indicates the U.S. EPA oral reference dose (RfD) for iAs (0.06 µg/kg-day).

**Figure 4 toxics-14-00013-f004:**
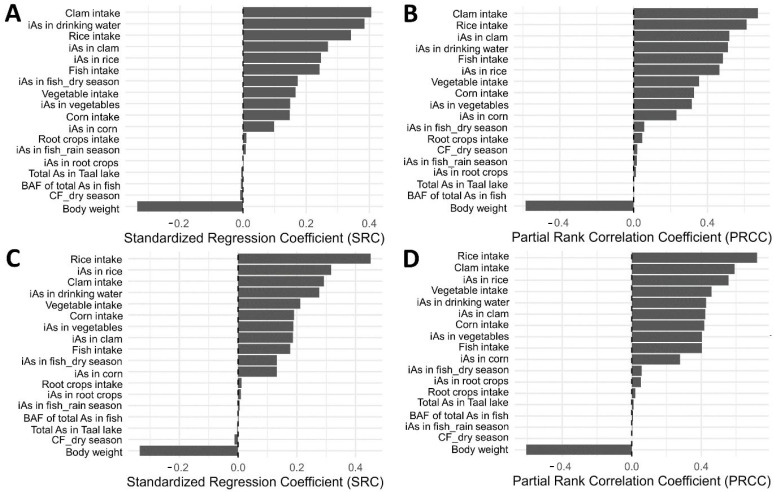
Sensitivity analysis of key input variables influencing estimated iAs exposure. Panels show standardized regression coefficients (SRC); lower-bound scenario in (**A**) and upper-bound scenario in (**C**) and partial rank correlation coefficients (PRCC); lower-bound in (**B**) and upper-bound in (**D**).

**Figure 5 toxics-14-00013-f005:**
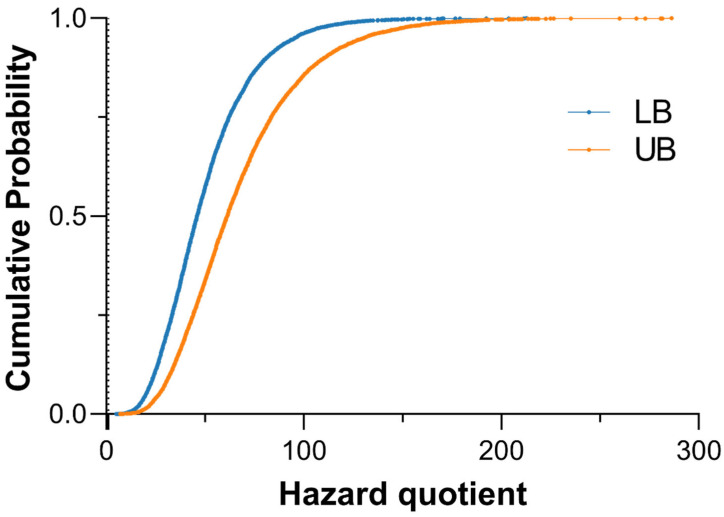
Cumulative distribution of hazard quotient (HQ) for total iAs intake under lower- (blue) and upper-bound (orange) exposure scenarios. The vertical line at HQ = 1 indicates the U.S. EPA reference dose threshold (0.06 µg/kg bw/day).

**Figure 6 toxics-14-00013-f006:**
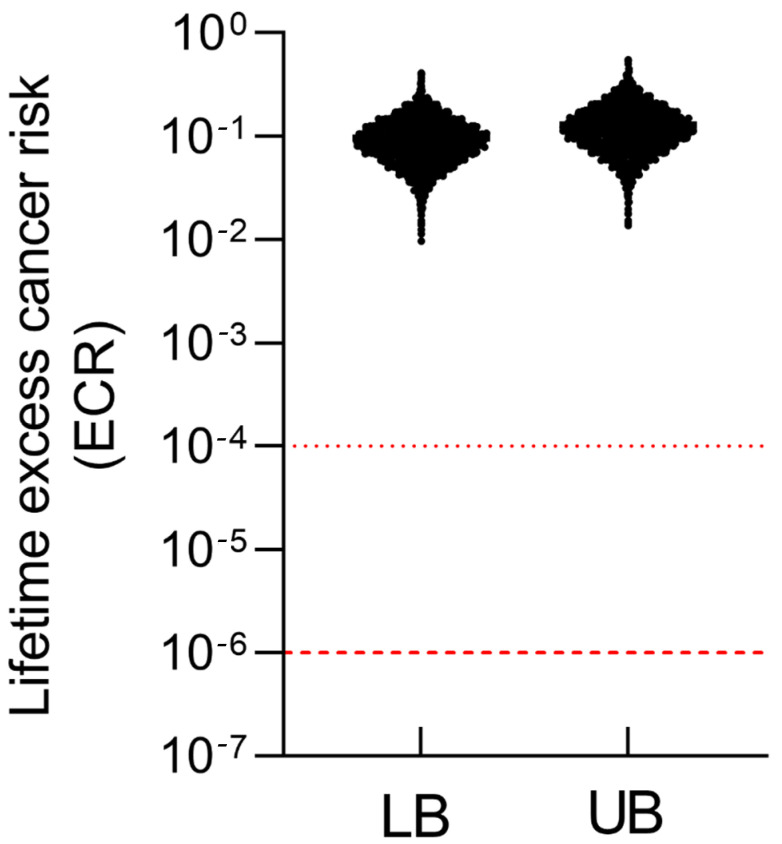
Lifetime excess cancer risk (ECR) under lower-bound (LB) and upper-bound (UB) exposure scenarios. Red dashed lines denote the U.S. EPA benchmark risk range for lifetime cancer risk (10^−6^–10^−4^).

**Table 1 toxics-14-00013-t001:** Environmental arsenic concentrations used as input parameters.

Medium	Parameter	Value/Range	Units
Drinking water	Total As	0.000735–0.11	mg/L
Soil	Total As	1.92–7.91	mg/kg
Lake water	Total As	0.005–0.0097	mg/L
Clam	Total As (dry weight)	5.0–6.3	mg/kg
Fish	Total As (derived)	From lake water × BAF	mg/kg

**Table 2 toxics-14-00013-t002:** Soil-to-crop transfer factor used in this study.

Crop	Soil-to-Crop Transfer Factor	Source
Rice	0.006–0.036	[32]
Corn	0.005–0.027	[33]
Vegetable	0.0003–0.028	[32]
Root crops	0.0028–0.007	[32]

**Table 3 toxics-14-00013-t003:** Food consumption parameters for the Monte Carlo simulation. Mean intake and standard deviation (SD) values were derived from national dietary references.

iAs Exposure Source	Mean (g/Day)	SD	Range (g/Day)
Fish	59	488.8	0–1525.4
Clam	8.5	70.4	0–219.8
Rice	263	872.8	0–2881.4
Corn	6	541.2	0–1629.6
Vegetables	58	680.8	0–2100.5
Root crops	7	104.7	0–321.1
Drinking water	1.791 (L/day)	–	Fixed

**Table 4 toxics-14-00013-t004:** Sex-specific anthropometric parameters for the Monte Carlo simulation.

Sex	Parameter	Mean	SD	Min	Max
Male	Body weight (kg)	61.3	9.0	40	155
	Height (cm)	163.0	6.5	60	220
Female	Body weight (kg)	54.3	8.5	35	145
	Height (cm)	154.0	6.0	55	210

## Data Availability

The original contributions presented in this study are included in the article/Appendix A. Further inquiries can be directed to the corresponding authors.

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
