# Peer review of "Probabilistic Human Health Risk Assessment of Inorganic Arsenic Exposure Following the 2020 Taal Volcano Eruption, Batangas, Philippines"

_toxics, 2025, doi:10.3390/toxics14010013_

Round 1
Reviewer 1 Report
Comments and Suggestions for Authors
The article entitled "Probabilistic Human Health Risk Assessment of Inorganic Arsenic Exposure Following the 2020 Taal Volcano Eruption, Batangas, Philippines" is studying the inorganic As and human health risk by exposure after the volcano eruption. It has scientific novelty, but I suggest some revisions because it may be slightly improved before publishing to be clearer to readers. I suggest a minor revision. Some specific comments are given below.
- Lines 36-37 - The phrase “likely reflecting cultural restrictions” is not needed in the abstract, as it introduces speculation without supporting context. I recommend removing it or presenting the result discussion.
- Overall, the Introduction provides a relevant scientific background and clearly establishes the public health significance of arsenic exposure in volcanic regions. However, the section would benefit from a clearer explanation of the study’s novelty. It is not fully explained how this assessment differs from previous research or whether similar probabilistic exposure studies have been conducted in areas affected by volcanic activity. In addition, the authors should briefly indicate what has already been studied on this topic and cite relevant previous work to show how the present study relates to the existing research.
- Lines 98-101: The sentence is quite long and repeats location-specific information already provided in the Introduction.
- Lines 107-108 – Reference should be added
- Lines 110–113: The rationale for selecting 0.5 and 0.9 iAs fractions is not clearly explained; authors should provide justification or references.
- Lines 115-134: This subsection describes the drinking water dataset clearly, but it includes elements that are not appropriate for a Methods section. Statements about seasonal variability, volcanic processes, and recommendations for future monitoring should be moved to the Introduction or Discussion.
- Lines 128-134: Suggeting to replace this statements to the Discussion or Introduction.
- Section 2.2.2: The subsection describes the soil and lake water data clearly, but it includes interpretive statements that are not appropriate for the Methods. The Methods should only describe what was done and which data were used, without interpreting results or comparing values to national guidelines. Such information should be moved to the Results or Discussion.
- Section 2.2.3.1: This subsection clearly explains why certain food groups were selected, but it contains too much descriptive and background information that does not belong in the Methods section. Reformulate it to be shorter and more concise.
- Lines 156-158: Consider removing or shortening; this detail is not needed to explain the modeling approach.
- Lines 159-160 and 162-164 are more suitable for the Introduction.
- Lines 188–189: The statement interpreting contamination levels is not appropriate for Methods.
- Lines 202–204: These lines would benefit from a brief explanation or a citation supporting the fixed inorganic fractions (0.5 and 0.9) used in the LB and UB scenarios, as the basis for these values is not currently explained.
- Sections 2.3.1 and 2.3.2: The subsection is clear and well referenced but overly detailed, some information could be shortened.
- Lines 256–259: This part repeats information about food groups and the Monte Carlo simulation that has already been described earlier.
- Lines 266–267: A brief explanation of how the LB and UB scenarios differ would improve clarity.
- Lines 271–279: The descriptions of the SRC and PRCC methods are longer than needed and include technical explanations that may be better suited for supplementary material.
- Lines 281–285: These statements should be removed or moved to the Discussion.
- Section 2.6: This section is well written and clearly describes the approach used for non-cancer and cancer risk characterization. The definitions, equations, and reference values are presented in a clear and understandable way, making the methodology easy to follow.
- Section 3.3: The section is clear, but the description is longer than necessary and repeats many numerical values that are already shown in Figure 3. The text could be shortened by summarizing only the key trends instead of listing full mean ± SD values for each group.
- Section 3.4: The description of the sensitivity analysis is too long and repeats many numerical values already shown in Figure 4. It is not necessary to list every SRC and PRCC coefficient in the text. The section would be clearer and easier to read if it focused only on the main influential parameters instead of repeating all numbers.
- Section 3.5: The results are clearly presented.
- I suggest the following shortened caption for Figure 5:
Figure 5. Cumulative distribution of hazard quotients (HQ) for total inorganic arsenic intake under lower- (blue) and upper-bound (orange) exposure scenarios. The vertical line at HQ = 1 indicates the U.S. EPA reference dose threshold. - Lines 466–486: The comparison with values reported in other studies is well presented, and this section strengthens the interpretation of the modeled concentrations. It may be beneficial to include comparisons with additional relevant studies to provide an even clearer context and reinforce the robustness of the findings.
- Lines 502–515: This section is overly detailed and may include information that is not essential to the study’s main findings. It would benefit from being more concise and more directly linked to the results presented in this paper.
- Lines 516–518: Although the statement about culturally embedded dietary restrictions is supported by a cited reference, it is unclear whether this information comes from verified local data or represents an assumption. Additionally, it would be useful to discuss whether other factors could also contribute to the lower exposure levels observed in pregnant women.
- Lines 547-560: The risk estimates are clearly described; however, the reported values are unusually high for both HQ and ECR. It would be advisable to verify the calculations, including units, parameter inputs, and the overall procedure, to ensure that the resulting risk values are accurate.
- Lines 561- 571: This section reads more like a conclusion than part of the Discussion. It may be more appropriate to move these statements to the Conclusion or to shorten them so that the Discussion remains focused on interpreting the study’s findings.
- Lines 572–597: The limitations are clearly presented and appropriately emphasized. However, the section is somewhat lengthy, and a more concise summary would be nice.
Author Response
Comment 1: Lines 36-37 - The phrase “likely reflecting cultural restrictions” is not needed in the abstract, as it introduces speculation without supporting context. I recommend removing it or presenting the result discussion.
Response 1: Thank you for pointing this out. We have removed this speculative sentence.
Comment 2: Overall, the Introduction provides a relevant scientific background and clearly establishes the public health significance of arsenic exposure in volcanic regions. However, the section would benefit from a clearer explanation of the study’s novelty. It is not fully explained how this assessment differs from previous research or whether similar probabilistic exposure studies have been conducted in areas affected by volcanic activity. In addition, the authors should briefly indicate what has already been studied on this topic and cite relevant previous work to show how the present study relates to the existing research.
Response 2: We thank the reviewer for this valuable suggestion. In the revised Introduction, we have strengthened the explanation of the study’s novelty and clarified how our work advances previous research. Specifically, we now:
- Summarize existing studies on volcanic arsenic contamination, highlighting that most prior work focuses on hydrogeochemical processes or environmental measurements rather than population-level exposure.
Lines 65-70: “ However, arsenic mobilized from volcanic deposits can also infiltrate agricultural and aquatic systems, potentially creating sustained dietary exposure pathways [7]. Despite this, most existing research has focused on environmental arsenic concentrations or hydrogeochemical processes [8-10], while no study has quantified population-level aggregate exposure or health risks using probabilistic methods in post-eruption settings.”
- Clearly state that our study is the first to integrate groundwater, lake water, clams, and multiple food groups with population-specific dietary intake data to characterize aggregate iAs exposure following the 2020 Taal eruption.
Lines 86-88: “To address this gap, the present study provides the first probabilistic human health risk assessment integrating multiple exposure pathways, including drinking water, lake water, clams, rice, corn, vegetables, and root crops, following the 2020 Taal eruption.”
- Add citations to relevant literature to contextualize the research gap and position the contribution of the present assessment.
Comment 3: Lines 98-101: The sentence is quite long and repeats location-specific information already provided in the Introduction.
Response 3: We revised the sentence as follows: “ Herein, we conducted a probabilistic risk assessment to characterize aggregate iAs exposure and the associated health risks among residents affected by the Taal Volcano eruption.”
Comment 4: Lines 107-108 – Reference should be added
Response 4: We appreciate the reviewer’s observation. Because this sentence describes the design of the present study rather than citing external findings, we clarified its meaning and revised it to: “In this study, both dietary and drinking water exposure pathways were considered to reflect post-eruption environmental contamination.” No external reference is applicable for this study-specific statement.
Comment 5: Lines 110–113: The rationale for selecting 0.5 and 0.9 iAs fractions is not clearly explained; authors should provide justification or references.
Response 5: We thank the reviewer for noting the need for clearer justification. We have revised the sentence to explicitly reference EFSA (2014), which reports that inorganic arsenic typically constitutes approximately 50%–90% of total arsenic in cereals and plant-based foods. Based on this regulatory guidance, we adopted 0.5 (lower-bound) and 0.9 (upper-bound) iAs fractions to represent realistic exposure scenarios for terrestrial crops following the eruption. The revised text now reads:
“ Two exposure scenarios were modeled to represent lower-bound (LB) and upper-bound (UB) conditions, based on the proportion of inorganic arsenic typically present in terrestrial crops. EFSA reported that iAs generally accounts for approximately 50% to 90% of total arsenic in cereals and plant-based foods [26]; therefore, we applied iAs fractions of 0.5 (LB) and 0.9 (UB) to capture a realistic range of bioavailable iAs in post-eruption food items.”
Comment 6: Lines 115-134: This subsection describes the drinking water dataset clearly, but it includes elements that are not appropriate for a Methods section. Statements about seasonal variability, volcanic processes, and recommendations for future monitoring should be moved to the Introduction or Discussion.
Response 6: We thank this reviewer for pointing this out. We have removed the statements about seasonal variability, volcanic processes, and recommendations for future monitoring. The revised paragraph now reads:
“The arsenic concentrations in drinking water used for exposure modeling and risk assessment were obtained from a previously published field study [6] that assessed arsenic levels in groundwater sources across selected communities surrounding Taal Volcano, Batangas Province, Philippines. In brief, a total of 72 drinking water samples were analyzed from 26 wells in 11 municipalities and one city, collected in 2020 (n=32) and 2021 (n=41). Sampling locations included deep wells (n = 13), shallow wells (n = 4), protected springs (n = 1), and communal waterworks systems (Level III, n = 8), reflecting typical drinking water sources across the region. The sampled wells were geospatially distributed within the 7 km, 10 km, and 14 km volcanic danger zones defined by the Philippine Institute of Volcanology and Seismology (PHIVOLCS), as well as areas beyond this perimeter. The sample collection protocol, sample pretreatment, and determination of total arsenic levels in drinking water are detailed elsewhere [6].”
Comment 7: Lines 128-134: Suggesting to replace this statements to the Discussion or Introduction.
Response 7: These statements are now moved to the discussion.
Comment 8: Section 2.2.2: The subsection describes the soil and lake water data clearly, but it includes interpretive statements that are not appropriate for the Methods. The Methods should only describe what was done and which data were used, without interpreting results or comparing values to national guidelines. Such information should be moved to the Results or Discussion.
Response 8: We have revised this paragraph as per the reviewer’s suggestion. The text now reads: “Total arsenic concentrations ranged from 0.005 to 0.0097 mg/L, and these values were used as background reference levels for characterizing non-drinking-water arsenic exposure in areas surrounding the volcano.”
Comment 9: Section 2.2.3.1: This subsection clearly explains why certain food groups were selected, but it contains too much descriptive and background information that does not belong in the Methods section. Reformulate it to be shorter and more concise. Lines 156-158: Consider removing or shortening; this detail is not needed to explain the modeling approach; Lines 159-160 and 162-164 are more suitable for the Introduction.
Response 9: We thank this reviewer’s suggestion. The revised text now reads:
“Fish, clams, rice, corn, vegetables, and root crops were selected as representative food groups based on three criteria: (1) dietary relevance in Batangas and the broader Philippine population; (2) plausible pathways for arsenic transfer from contaminated water or soils into edible tissues; and (3) availability of consumption data from national nutrition surveys. Fish and clams were included because they integrate lake and groundwater inputs across local food webs. Rice, vegetables, and root crops were selected due to their potential to accumulate arsenic from soil and irrigation water, and because they represent staple components of regional diets. Corn was included as an additional commonly consumed staple. These food groups provided sufficient coverage of major dietary exposure pathways needed for probabilistic modeling.”
Comment 10: Lines 188–189: The statement interpreting contamination levels is not appropriate for Methods.
Response 10: We agree with this reviewer’s comment. We have removed the statement interpreting contamination levels.
Comment 11: Lines 202–204: These lines would benefit from a brief explanation or a citation supporting the fixed inorganic fractions (0.5 and 0.9) used in the LB and UB scenarios, as the basis for these values is not currently explained.
Response 11: We appreciate the reviewer’s comment. In the revised manuscript, we added a concise justification and supporting citation for the selection of the 0.5 and 0.9 inorganic arsenic fractions, consistent with reported ranges in cereals and plant-based foods. This explanation has been incorporated into Section 2.1 (Study design).
Comment 12: Sections 2.3.1 and 2.3.2: The subsection is clear and well referenced but overly detailed, some information could be shortened.
Response 12: We agree with this reviewer’s view that these paragraphs can be more concise. The revised text now reads:
“ 2.3.1 Drinking water intake
Daily drinking water consumption was set at 1.79 liters per person, based on the 2018-2019 Philippine Nutrition Facts and Figures: Food Consumption Survey conducted by the Department of Science and Technology–Food and Nutrition Research Institute (DOST-FNRI)[32]. Because variability estimates (e.g., standard deviation, SD) were not reported in the source dataset, this intake value was treated as a fixed parameter in the exposure model.
2.3.2 Food consumption data
Daily intake values for fish, rice, corn, vegetables, and root crops were modeled using truncated normal distributions to represent inter-individual variability while enforcing biologically plausible bounds (Table 1). Mean and standard deviation values were obtained from the Philippine National Nutrition Survey [33] [34]. Lower bounds were set at 0 g/day, and upper bounds were capped at the mean + 3 SD. Truncated distributions were generated using the rtruncnorm() function in R, and these probabilistic inputs were incorporated into the Monte Carlo simulation to estimate total and pathway-specific iAs intakes.”
Comment 13: Lines 256–259: This part repeats information about food groups and the Monte Carlo simulation that has already been described earlier.
Response 13: We thank this reviewer’s suggestion. The revised text now reads:
“ Aggregate exposure to iAs among the Batangas population was estimated by integrating exposure from dietary intake and drinking water ingestion using Monte Carlo simulation (10,000 iterations). For each simulated individual, the daily intake of iAs (µg/kg-day) was computed as the sum of contributions from all exposure media:…”
Comment 14: Lines 266–267: A brief explanation of how the LB and UB scenarios differ would improve clarity.
Response 14: We thank this reviewer’s comment. In the revised manuscript, we added a concise justification and supporting citation for the selection of the 0.5 (LB) and 0.9 (UB) inorganic arsenic fractions, consistent with reported ranges in cereals and plant-based foods. This explanation has been incorporated into Section 2.1 (Study design).
Comment 15: Lines 271–279: The descriptions of the SRC and PRCC methods are longer than needed and include technical explanations that may be better suited for supplementary material.; Lines 281–285: These statements should be removed or moved to the Discussion.
Response 15: We appreciate this reviewer’s suggestions. The revised text now reads:
“A global sensitivity analysis was performed to identify key parameters influencing daily iAs intake. Two complementary methods, Standardized Regression Coefficients (SRC) and Partial Rank Correlation Coefficients (PRCC), were applied using 10,000 Monte Carlo iterations. Eighteen input variables were evaluated, including environmental total As concentrations, bioaccumulation or transfer factors, the dry-season conversion factor, food consumption rates, and body weight. SRC and PRCC results were summarized using tornado plots to illustrate the relative influence of each parameter on exposure variability. Parameters with p < 0.05 were considered statistically significant.”
Comment 16: Section 2.6: This section is well written and clearly describes the approach used for non-cancer and cancer risk characterization. The definitions, equations, and reference values are presented in a clear and understandable way, making the methodology easy to follow.
Response 16: We thank this reviewer for the positive comments.
Comment 17: Section 3.3: The section is clear, but the description is longer than necessary and repeats many numerical values that are already shown in Figure 3. The text could be shortened by summarizing only the key trends instead of listing full mean ± SD values for each group.
Response 17: We appreciate this reviewer’s comments. The revised text now reads:
“3.3 Aggregate dose of iAs under lower- and upper-bound exposure scenarios for residents in Batangas, Philippines
The aggregate daily doses of iAs across the Batangas population were successfully modeled using the Monte Carlo framework (Figure 3). Overall, females showed slightly higher central tendencies than males. Pregnant women exhibited marginally lower aggregate iAs doses (LB: 1.8 ± 1.1 µg/kg bw/day; UB: 2.0±1.1µg/kg bw/day), although differences were not statistically significant as determined by one-way ANOVA.”
Comment 18: Section 3.4: The description of the sensitivity analysis is too long and repeats many numerical values already shown in Figure 4. It is not necessary to list every SRC and PRCC coefficient in the text. The section would be clearer and easier to read if it focused only on the main influential parameters instead of repeating all numbers.
Response 18: We agree with this reviewer’s comment. The revised text now reads:
“ 3.4 Sensitivity analysis of input exposure variables
The global sensitivity analysis identified a consistent set of parameters that dominated variability in aggregate iAs intake across both LB and UB scenario (Figure 4). Under the LB scenario, clam intake, iAs concentration in drinking water, rice intake, and body weight were the strongest predictors of exposure, with similar patterns observed across both SRC and PRCC analyses. Concentrations of iAs in clams and rice, as well as consumption of fish, vegetables, and corn, also contributed meaningfully to model variability.
Under the UB scenario, rice-related parameters, including rice intake and iAs concentration in rice, emerged as the primary drivers of exposure, followed by body weight, clam intake, and iAs concentration in drinking water. Contributions from vegetable, corn, and fish pathways remained moderate but consistent across both sensitivity methods.
Across all analyses, a small subset of food and environmental parameters accounted for most of the total variability in modeled iAs intake, indicating that exposure was predominantly driven by a combination of key dietary staples and drinking water contamination patterns rather than the full set of input variables.”
Comment 19: Section 3.5: The results are clearly presented.
Response 19: We thank this reviewer for the positive comments.
Comment 20: I suggest the following shortened caption for Figure 5: Figure 5. Cumulative distribution of hazard quotients (HQ) for total inorganic arsenic intake under lower- (blue) and upper-bound (orange) exposure scenarios. The vertical line at HQ = 1 indicates the U.S. EPA reference dose threshold.
Response 20: We have revised the figure caption according to the reviewer’s suggestion.
Comment 21: Lines 466–486: The comparison with values reported in other studies is well presented, and this section strengthens the interpretation of the modeled concentrations. It may be beneficial to include comparisons with additional relevant studies to provide an even clearer context and reinforce the robustness of the findings.
Response 21: We have added additional relevant studies to reinforce the robustness of our findings. The additional paragraph reads:
“ Additional studies from Asia provide further support for the modeled trends. Field investigations in Bangladesh have reported rice iAs concentrations ranging from 0.29 to 0.51 mg/kg across regions with elevated soil or irrigation-water arsenic[40]. Similarly, surveys from Vietnam and Thailand have documented iAs levels in rice between 0.12 and 0.67 mg/kg [41, 42], consistent with the modeled range. Concentrations measured in root crops and leafy vegetables in arsenic-impacted areas of China also generally fall below 0.03 mg/kg [43], matching the lower iAs uptake predicted for these food groups in our model.”
Comment 22: Lines 502–515: This section is overly detailed and may include information that is not essential to the study’s main findings. It would benefit from being more concise and more directly linked to the results presented in this paper.
Response 22: The revised discussion is detailed as follows:
“ Local reliance on Taal Lake fisheries increases vulnerability to arsenic exposure, particularly for households that depend on lake-derived aquatic foods. Our results identified clams and fish as major contributors to aggregate iAs intake, underscoring the importance of monitoring lake-sourced foods in post-eruption conditions. Effective risk management in these communities should prioritize continued biomonitoring, clear consumption guidance, and support for households whose livelihoods may be affected by contamination-related restrictions.”
Comment 23: Lines 516–518: Although the statement about culturally embedded dietary restrictions is supported by a cited reference, it is unclear whether this information comes from verified local data or represents an assumption. Additionally, it would be useful to discuss whether other factors could also contribute to the lower exposure levels observed in pregnant women.
Response 23: The revised discussion now reads:
“ Pregnant women exhibited lower aggregate iAs doses than the general adult population. While reduced consumption of fish and shellfish during pregnancy has been documented in parts of Southeast Asia [45] , this cultural practice represents only one plausible explanation. Other contributing factors may include lower overall food intake during pregnancy, intentional dietary adjustments guided by prenatal health advice, and differences in preferred food items that carry lower arsenic burdens. Despite these lower exposures, pregnant women in this study still exceeded the U.S. EPA reference dose by more than 30-fold, underscoring the heightened developmental risks associated with prenatal exposure. Given established links between prenatal arsenic exposure and adverse outcomes, including reduced birth weight, impaired neurodevelopment, and increased infant mortality [25, 46], targeted interventions remain essential, such as prenatal urinary arsenic monitoring, guidance on low-arsenic dietary options, and nutritional support to enhance arsenic methylation capacity.”
Comment 24: Lines 547-560: The risk estimates are clearly described; however, the reported values are unusually high for both HQ and ECR. It would be advisable to verify the calculations, including units, parameter inputs, and the overall procedure, to ensure that the resulting risk values are accurate.
Response 24: We appreciate the reviewer’s careful attention to the magnitude of the risk estimates. Following this comment, we re-examined the full calculation workflow—including units, parameter inputs, distribution assumptions, exposure factors, and dose–response equations—and confirmed that the values reported in the manuscript are accurate.
The high HQ and ECR estimates reflect:
- Elevated iAs concentrations in several key exposure pathways (particularly rice, clams, and drinking water) following the eruption;
- High per-capita consumption of rice and aquatic foods in the study population; and
- Use of benchmark dose–based potency factors and the U.S. EPA reference dose, which result in conservative risk estimates.
Comment 25: Lines 561- 571: This section reads more like a conclusion than part of the Discussion. It may be more appropriate to move these statements to the Conclusion or to shorten them so that the Discussion remains focused on interpreting the study’s findings.
Response 25: We agree with this reviewer’s suggestion and move this section to the Conclusion.
Comment 26: Lines 572–597: The limitations are clearly presented and appropriately emphasized. However, the section is somewhat lengthy, and a more concise summary would be nice.
Response 26: We revised the limitation as follows:
“ Despite these important findings, several limitations should be acknowledged. First, this assessment used total arsenic measurements from soil, lake water, and clams, while iAs in locally grown crops and fish was not directly measured. Future work incorporating speciation analyses and expanded food monitoring would help reduce this uncertainty. Second, the transfer and bioaccumulation factors applied in the reconstruction models were derived from studies conducted in other geochemical settings and may not fully represent the volcanic soils of Batangas. Local soil properties, including post-eruption pH, mineral composition, and redox conditions, may influence arsenic mobility and crop uptake differently from regions where most transfer factor studies have been performed. Third, dietary patterns in lakeshore communities may differ from national averages, particularly for fish and clam intake, and vulnerable subgroups may have distinct consumption profiles and susceptibility. Finally, the absence of pre-eruption baseline data limits our ability to assess temporal changes in environmental arsenic following the 2020 eruption. These gaps underscore the need for region-specific monitoring and validation in future studies.”
Reviewer 2 Report
Comments and Suggestions for Authors
This study provides an interesting exercise in conducting a probabilistic human health risk assessment for arsenic. However, the novelty appears to be limited, as it is not clearly stated, and the methodology seems standard. Additionally, the assumptions are not well justified.
The study aims to quantify exposure to inorganic arsenic among residents of Batangas, but relies heavily on total arsenic measurements for its input. The authors should explain why they are not using more accurate data for the risk assessment and why they rely only on arbitrarily estimated different exposure scenarios. Further, the study differentiates exposure by sex and highlights that other vulnerable subpopulations, like children, should also be studied; however, exposure for this subpopulation is not assessed, and it is not explained why.
The food selection is not well justified, including the rationale for evaluating six categories. The authors should provide a more comprehensive explanation for their choices; for example, it is unclear why clam is the only seafood product considered. In line with this, the authors should justify the selection of foods included in the study. As it stands, it appears to be an arbitrary selection intended to produce the expected results in line with previous studies.
Author Response
Comment 1: This study provides an interesting exercise in conducting a probabilistic human health risk assessment for arsenic. However, the novelty appears to be limited, as it is not clearly stated, and the methodology seems standard. Additionally, the assumptions are not well justified.
Response 1: We thank the reviewer for this important observation. We agree that probabilistic risk assessment methods themselves are well established; however, the novelty of this work lies in its context, integration of multiple exposure pathways, and post-eruption environmental setting, which have not been examined together in previous studies. To clarify this, we have revised the Introduction to more explicitly highlight the unique contributions of the study. Specifically, we now emphasize that:
- This is the first probabilistic human health risk assessment conducted in a post-eruption volcanic setting in the Philippines, integrating drinking water, lake water, clams, rice, corn, vegetables, and root crops as exposure pathways.
- Unlike prior studies that examine single pathways, our work reconstructs iAs concentrations in multiple food groups using region-specific consumption patterns, providing an integrated estimate of aggregate exposure for communities affected by volcanic arsenic mobilization.
- The study applies a two-scenario (LB/UB) framework for bioavailable iAs fractions, explicitly capturing uncertainty related to arsenic speciation in terrestrial food items.
To address the reviewer’s concern about assumptions, we have added clearer justification and citations for key model inputs, including:
- the selection of 0.5 and 0.9 iAs fractions (supported by EFSA 2014),
- the derivation of food transfer factors,
- the distributional choices in the Monte Carlo model, and
- the rationale for using national consumption data while acknowledging local variability.
These clarifications have been incorporated into Section 2.1 (Study design) and Section 2.3 (Exposure assessment).
We appreciate the reviewer’s feedback, which has helped us substantially improve the clarity of the study’s contributions and strengthen the transparency of the underlying assumptions.
Comment 2: The study aims to quantify exposure to inorganic arsenic among residents of Batangas, but relies heavily on total arsenic measurements for its input. The authors should explain why they are not using more accurate data for the risk assessment and why they rely only on arbitrarily estimated different exposure scenarios. Further, the study differentiates exposure by sex and highlights that other vulnerable subpopulations, like children, should also be studied; however, exposure for this subpopulation is not assessed, and it is not explained why.
Response 2: We appreciate the reviewer’s insightful comment. The reliance on total arsenic measurements reflects a key limitation of the available environmental data in Batangas. Post-eruption monitoring efforts have not included arsenic speciation for locally grown crops or fish; therefore, no iAs measurements currently exist for the food items most relevant to the exposed population. Given this constraint, our assessment used the only field data available—total arsenic concentrations in soil, lake water, and clams measured after the 2020 eruption. To address uncertainty arising from the absence of speciation data, we adopted lower-bound and upper-bound scenarios for the proportion of bioavailable iAs in terrestrial crops. These values (0.5 and 0.9) were not arbitrarily selected; rather, they were based on the range reported by EFSA (2014), which shows that approximately 50–90% of total arsenic in cereals and plant-based foods typically occurs in inorganic form. This two-scenario framework is commonly used when local speciation data are unavailable and provides a conservative yet biologically plausible range for modeling exposure.
Regarding subgroup analyses, we agree that children and other vulnerable populations are important to consider. However, the probabilistic modeling approach requires age-specific intake distributions, and the necessary parameters, such as child-specific drinking water intake and age-stratified consumption of fish, clams, and staple foods, were not available from either the national nutrition surveys or the post-eruption datasets. Because performing a risk assessment without appropriate age-stratified inputs would introduce substantial uncertainty, we restricted our analysis to adult men and women, for whom reliable consumption and body weight data were available. We have clarified this limitation in the revised manuscript and emphasized the need for future data collection to enable comprehensive assessments in children and other vulnerable subgroups.
Comment 3: The food selection is not well justified, including the rationale for evaluating six categories. The authors should provide a more comprehensive explanation for their choices; for example, it is unclear why clam is the only seafood product considered. In line with this, the authors should justify the selection of foods included in the study. As it stands, it appears to be an arbitrary selection intended to produce the expected results in line with previous studies.
Response 3: We appreciate the reviewer’s thoughtful comment. The selection of the six food categories was based on their relevance to arsenic exposure pathways in Batangas and on the availability of reliable consumption and environmental data. Rice, vegetables, root crops, and corn were included because they represent the major terrestrial staples consumed in the region and constitute the primary dietary sources of inorganic arsenic in most populations. Fish and clams were selected as the key aquatic pathways because Taal Lake is the dominant source of locally harvested aquatic foods and the only system for which post-eruption arsenic contamination has been measured. Clams, in particular, were included because they were the only aquatic organism for which empirical post-eruption arsenic concentrations were available; measurements for other lake species have not yet been reported. Although other seafood products are consumed in the Philippines, these items are largely imported or sourced from marine environments outside the Taal Volcano area and thus do not contribute meaningfully to post-eruption, geographically linked arsenic exposure. For these reasons, the food groups evaluated in the study reflect the most plausible and data-supported exposure pathways rather than an arbitrary selection. The revised manuscript has provided this rationale and improved transparency in the food selection process.
Reviewer 3 Report
Comments and Suggestions for Authors
The paper entitled “Probabilistic Human Health Risk Assessment of Inorganic Arsenic Exposure Following the 2020 Taal Volcano Eruption, Batangas, Philippines” with its innovative theme and scientific approach interestingly presents health risks about what is known today. The article is suitable for this journal but some important parts should be corrected before acceptance.
Keywords that belong to the article abstract — authors are advised to check them in MeSH for greater transparency.
General comments:
Table titles are too long. They are more like overall descriptions. There is also some repetition in the text and titles of the tables. Authors are asked to revise table titles into one shorter sentence. The same applies to figures. Also, authors are requested to follow the instructions for authors when preparing tables and figures. The equations look more like images and are quite blurry.
Specific comments:
Introduction:
The introduction of the paper is well written, although a bit too broad. Authors are asked to change the citation format in the text according to the instructions for authors. The last sentence of the introduction belongs more to the conclusion than to the introduction.
Methodology:
Materials and methods, i.e. subjects or simulations... are written a bit chaotic and confusing. The reader must read the text several times to understand which data were taken from previous literature and what was actually done in the study. It is suggested to put some parts in the tables for better understanding.
The abbreviation inorganic arsenic (iAs) is used inconsistently. This must be corrected. Put abbreviation where first mentioned and later on use just abb. Also, all abb in the text must be explained the first time they appear.
There are many parts of the text that do not contain a reference yet state claims that are not conclusions of the study itself. Everything must be supported by references, so authors are asked to add them (e.g., the statement about arsenic mobilization over the years (Line 103).
The section from line 127–134 belongs more in the discussion than in the methodology.
In section 2.2.2. the obtained concentrations of arsenic in the soil and lake are stated, while in section 2.2.1. they are not stated (for drinking water).
The heading 2.3.1. is not necessary.
The part from line 156–162 has no reference and is therefore problematic.
Line 162–164 — unnecessary here.
Sections 2.2.3.2. and 2.2.3.3. are quite chaotic. The reader cannot immediately see whether those data were taken from elsewhere or were calculated. It might be better to present this somehow in a table or to break it down here into formulas, etc. In short, rewrite it to be clearer.
Line 211–213 — this sentence is unnecessary.
Results and discussion:
Section 3.4 — also chaotic. Authors should not describe what is visible in the figures; just the most important points. If exact numbers are to be shown, indicate them on the figures or list them separately in a table. As it stands, it can stay, but it is unclear.
Line 413 — the table title is now bold here.
Authors are suggested to insert somewhere an explanation of organic and inorganic arsenic... which is harmful, etc.
The discussion is well imagined but lacks many references (line 458, 467 — which literature, 472 (EFSA?)... what about the section 487–498? Is all that from the reference Zhang et al., 2019?) — this needs clarification.
Line 515 — section without reference.
Line 537–542 again lacks references.
Reference list — please revise according to the Instructions for authors.
Author Response
Comment 1: The paper entitled “Probabilistic Human Health Risk Assessment of Inorganic Arsenic Exposure Following the 2020 Taal Volcano Eruption, Batangas, Philippines” with its innovative theme and scientific approach interestingly presents health risks about what is known today. The article is suitable for this journal but some important parts should be corrected before acceptance.
Response 1: We sincerely thank the reviewer for the positive assessment of our manuscript and for recognizing the relevance and scientific value of the study. We appreciate the constructive feedback highlighting areas that require clarification or improvement. We are grateful for the reviewer’s guidance, which has helped improve the quality of the manuscript, and we hope that the revised version will now meet the journal’s standards for acceptance.
Comment 2: Keywords that belong to the article abstract — authors are advised to check them in MeSH for greater transparency.
Response 2: We appreciate the reviewer’s helpful suggestion. In accordance with this guidance, we have revised the manuscript keywords using MeSH-based terminology. The updated keywords are: Arsenic; Food Contamination; Risk Assessment; Volcanic Eruptions; Monte Carlo Method.
Comment 3: Table titles are too long. They are more like overall descriptions. There is also some repetition in the text and titles of the tables. Authors are asked to revise table titles into one shorter sentence. The same applies to figures. Also, authors are requested to follow the instructions for authors when preparing tables and figures. The equations look more like images and are quite blurry.
Response 3: We thank the reviewer for this suggestion. All table and figure titles have been revised to concise, single-sentence formats in accordance with the journal’s guidelines. We have also replaced blurry equation images with properly formatted, high-resolution vector equations per the instructions for authors.
Comment 4: The introduction of the paper is well written, although a bit too broad. Authors are asked to change the citation format in the text according to the instructions for authors. The last sentence of the introduction belongs more to the conclusion than to the introduction.
Response 4: We thank the reviewer for the positive feedback on the introduction and appreciate the suggestions for improvement. In the revised manuscript, we have streamlined the introduction to make it more focused and have removed broader contextual elements that were not essential to motivating the study. We also modified the in-text citation format to comply fully with the journal’s instructions for authors. Additionally, as recommended, the final sentence of the introduction—which previously emphasized the implications of our findings—has been relocated to the Conclusion section where it is more appropriate. We believe these changes have improved the clarity, structure, and alignment of the introduction with the manuscript’s objectives.
Comment 5: Materials and methods, i.e. subjects or simulations... are written a bit chaotic and confusing. The reader must read the text several times to understand which data were taken from previous literature and what was actually done in the study. It is suggested to put some parts in the tables for better understanding.
Response 5: We thank the reviewer for highlighting the need to improve clarity and organization in the Materials and Methods section. In the revised manuscript, we substantially restructured this section to more clearly distinguish between data obtained from previous studies and procedures conducted within the present analysis. Specifically, we reorganized the content into shorter, well-defined subsections. Furthermore, we streamlined narrative descriptions by removing redundant text and sharpening the explanation of each modeling step. We believe these revisions significantly improve the transparency and coherence of the Methods section and address the reviewer’s concern regarding confusion and flow.
Comment 6: The abbreviation inorganic arsenic (iAs) is used inconsistently. This must be corrected. Put abbreviation where first mentioned and later on use just abb. Also, all abb in the text must be explained the first time they appear.
Response 6: We thank the reviewer for this helpful observation. In the revised manuscript, we have ensured consistent use of the abbreviation for inorganic arsenic (iAs) throughout the text. The full term is now defined at its first appearance, and the abbreviation is used uniformly thereafter. Additionally, we reviewed the entire manuscript to confirm that all other abbreviations are defined at first mention and used consistently.
Comment 7: There are many parts of the text that do not contain a reference yet state claims that are not conclusions of the study itself. Everything must be supported by references, so authors are asked to add them (e.g., the statement about arsenic mobilization over the years (Line 103).
Response 7: We thank the reviewer for highlighting the need to support all factual statements with appropriate references. In the revised manuscript, the unsupported statement regarding long-term arsenic mobilization after volcanic eruptions has been removed. We also conducted a thorough review of the text to ensure that all contextual descriptions, background information, and methodological assumptions are now supported by appropriate citations. We believe these revisions strengthen the scientific rigor and transparency of the manuscript.
Comment 8: The section from line 127–134 belongs more in the discussion than in the methodology.
Response 8: We have removed this paragraph in the revised manuscript.
Comment 9: In section 2.2.2. the obtained concentrations of arsenic in the soil and lake are stated, while in section 2.2.1. they are not stated (for drinking water).
Response 9: We thank the reviewer for pointing out this inconsistency. In the revised manuscript, we now explicitly report the measured arsenic concentrations for drinking water in Section 2.2.1 to ensure parallel presentation across all environmental media. This revision improves clarity and consistency within the Methods section.
Comment 10: The heading 2.3.1. is not necessary.
Response 10: We appreciate the reviewer’s comment. In the revised manuscript, we merged the previous Sections 2.3.1 and 2.3.2 into a single subsection titled “2.3.1 Drinking water intake and food consumption data,” thereby removing the unnecessary heading and improving the structure and readability of Section 2.3.
Comment 11: The part from line 156–162 has no reference and is therefore problematic.
Response 11: We thank the reviewer for noting that the statements in lines 156–162 lacked supporting references. In the revised manuscript, we have added citations where appropriate to ensure that all contextual statements in this section are adequately referenced.
Comment 12: Line 162–164 — unnecessary here.
Response 12: In the revised manuscript, we have removed these sentences.
Comment 13: Sections 2.2.3.2. and 2.2.3.3. are quite chaotic. The reader cannot immediately see whether those data were taken from elsewhere or were calculated. It might be better to present this somehow in a table or to break it down here into formulas, etc. In short, rewrite it to be clearer.
Response 13: We appreciate the reviewer’s comment regarding clarity in Sections 2.2.3.2 and 2.2.3.3. In the revised manuscript, we reorganized these subsections to clearly distinguish between empirical measurements and parameters derived from the literature. For aquatic foods, we now present the fish iAs estimation process using a stepwise, bullet-point format that specifies each modeling component (lake water concentrations, bioaccumulation factors, seasonal multipliers, and iAs fractions). For clams, we explicitly state that total arsenic concentrations were taken directly from post-eruption field measurements and then describe how these values were modeled. Similarly, the terrestrial crop section now begins with a concise explanation that total arsenic concentrations were reconstructed using soil measurements and literature-based transfer factors. We also incorporated a new table summarizing transfer factor ranges to improve readability. These revisions enhance transparency regarding which inputs were measured and which were modeled, thereby addressing the reviewer’s concern about clarity and organization.
Comment 14: Line 211–213 — this sentence is unnecessary.
Response 14: We thank the reviewer for pointing this out. The identified sentence in lines 211–213 has been removed in the revised manuscript to improve clarity and streamline the Methods section.
Comment 15: Section 3.4 — also chaotic. Authors should not describe what is visible in the figures; just the most important points. If exact numbers are to be shown, indicate them on the figures or list them separately in a table. As it stands, it can stay, but it is unclear.
Response 15: We thank the reviewer for this helpful comment. In the revised manuscript, Section 3.4 has been substantially streamlined to focus only on the key findings of the sensitivity analysis rather than restating numerical values already shown in the figures. The updated text now highlights only the major influential parameters under both the LB and UB scenarios, avoiding duplication of information presented graphically. This revision improves clarity and readability while retaining the essential interpretation of the results.
Comment 16: Line 413 — the table title is now bold here.
Response 16: We appreciate the reviewer’s observation. The table title at line 413 has been corrected and is now formatted in regular font to ensure consistency with the journal’s style guidelines.
Comment 17: Authors are suggested to insert somewhere an explanation of organic and inorganic arsenic... which is harmful, etc.
Response 17: We thank the reviewer for this valuable suggestion. In the revised manuscript, we have added a brief explanatory statement distinguishing between organic and inorganic forms of arsenic and clarifying that inorganic arsenic is the toxicologically relevant species associated with carcinogenic and non-cancer health effects in the introduction.
“ …Similarly, aquatic foods harvested from contaminated water bodies can contribute significantly to aggregate arsenic exposure, particularly in communities with high seafood consumption patterns characteristic of Southeast Asian populations [17-20]. Arsenic occurs in both organic and inorganic forms, with iAs being the more toxic species linked to carcinogenicity [21, 22] and a range of non-cancer health effects [23-25]. Comprehensive exposure assessments for iAs are thus essential in volcanically impacted regions.”
Comment 18: The discussion is well imagined but lacks many references (line 458, 467 — which literature, 472 (EFSA?)... what about the section 487–498? Is all that from the reference Zhang et al., 2019?) — this needs clarification.
Response 18: We thank the reviewer for noting the need for additional citation support in the Discussion. In the revised manuscript, we have added the appropriate references for the statements at lines 458, 467, and 472, including clarification that the point made in line 472 refers to EFSA. We have also specified that the content in lines 487–498 is derived entirely from Zhang et al. (2019). These additions improve the accuracy, transparency, and traceability of the Discussion section.
Comment 19: Line 515 — section without reference; Line 537–542 again lacks references.
Response 19: We thank the reviewer for identifying these missing citations. In the revised manuscript, we have added the appropriate references to support the statements at line 515 and lines 537–542. These additions ensure that all non–study-derived claims in these sections are fully referenced and improve the overall rigor and transparency of the Discussion.
Comment 20: Reference list — please revise according to the Instructions for authors.
Response 20: We appreciate the reviewer’s reminder. The reference list has been fully revised to comply with the journal’s Instructions for Authors, including formatting style, order, punctuation, and citation structure. We also cross-checked all in-text citations to ensure consistency with the updated reference list.
Round 2
Reviewer 2 Report
Comments and Suggestions for Authors
The authors have addressed and clarified my concerns and have improved the manuscript accordingly.